# Potential Anti-Obesity, Anti-Steatosis, and Anti-Inflammatory Properties of Extracts from the Microalgae *Chlorella vulgaris* and *Chlorococcum amblystomatis* under Different Growth Conditions

**DOI:** 10.3390/md20010009

**Published:** 2021-12-22

**Authors:** Ana Regueiras, Álvaro Huguet, Tiago Conde, Daniela Couto, Pedro Domingues, Maria Rosário Domingues, Ana Margarida Costa, Joana Laranjeira da Silva, Vitor Vasconcelos, Ralph Urbatzka

**Affiliations:** 1Blue Biotechnology and Ecotoxicology Group, CIIMAR/CIMAR, Interdisciplinary Centre of Marine and Environmental Research, Terminal de Cruzeiros do Porto de Leixões, University of Porto, 4450-208 Matosinhos, Portugal; aregueiras@ciimar.up.pt (A.R.); alvarohl96@gmail.com (Á.H.); vmvascon@fc.up.pt (V.V.); 2Departamento de Biologia, Faculdade de Ciências, Universidade do Porto, Rua do Campo Alegre, Edifício FC4, 4169-007 Porto, Portugal; 3Mass Spectrometry Centre, LAQV-REQUIMTE, Department of Chemistry, University of Aveiro, Santiago University Campus, 3810-193 Aveiro, Portugal; tiagoalexandreconde@ua.pt (T.C.); danielacouto@ua.pt (D.C.); p.domingues@ua.pt (P.D.); mrd@ua.pt (M.R.D.); 4CESAM—Centre for Environmental and Marine Studies, Department of Chemistry, University of Aveiro, Santiago University Campus, 3810-193 Aveiro, Portugal; 5Allmicroalgae, R&D Department, Rua 25 de Abril, 2445-287 Pataias, Portugal; margarida.costa@niva.no (A.M.C.); joana.g.silva@allmicroalgae.com (J.L.d.S.)

**Keywords:** microalgae, anti-obesity, anti-inflammation, anti-steatosis, molecular networking

## Abstract

Microalgae are known as a producer of proteins and lipids, but also of valuable compounds for human health benefits (e.g., polyunsaturated fatty acids (PUFAs); minerals, vitamins, or other compounds). The overall objective of this research was to prospect novel products, such as nutraceuticals from microalgae, for application in human health, particularly for metabolic diseases. *Chlorella vulgaris* and *Chlorococcum amblystomatis* were grown autotrophically, and *C. vulgaris* was additionally grown heterotrophically. Microalgae biomass was extracted using organic solvents (dichloromethane, ethanol, ethanol with ultrasound-assisted extraction). Those extracts were evaluated for their bioactivities, toxicity, and metabolite profile. Some of the extracts reduced the neutral lipid content using the zebrafish larvae fat metabolism assay, reduced lipid accumulation in fatty-acid-overloaded HepG2 liver cells, or decreased the LPS-induced inflammation reaction in RAW264.7 macrophages. Toxicity was not observed in the MTT assay in vitro or by the appearance of lethality or malformations in zebrafish larvae in vivo. Differences in metabolite profiles of microalgae extracts obtained by UPLC-LC-MS/MS and GNPS analyses revealed unique compounds in the active extracts, whose majority did not have a match in mass spectrometry databases and could be potentially novel compounds. In conclusion, microalgae extracts demonstrated anti-obesity, anti-steatosis, and anti-inflammatory activities and could be valuable resources for developing future nutraceuticals. In particular, the ultrasound-assisted ethanolic extract of the heterotrophic *C. vulgaris* significantly enhanced the anti-obesity activity and demonstrated that the alteration of culture conditions is a valuable approach to increase the production of high-value compounds.

## 1. Introduction

Obesity, according to the World Health Organization (WHO), is defined as an abnormal or excessive fat accumulation that may impair health, reduce well-being, and increase morbidity and mortality. Nowadays, it is considered a global epidemic, leading to more than 2.8 million deaths per year. Obesity prevalence almost tripled in 40 years (1975–2016), and in 2016, the WHO estimated that 380 million children under 19 were overweight or obese, and up to 1.9 billion adults were overweight [1]. The excessive accumulation of fat tissue accompanied by mild chronic inflammation of the tissue defines obese individuals [2]. Obesity is correlated with an exponential increase of metabolic syndrome (MetS) [3], leading to type 2 diabetes, nonalcoholic fatty liver disease (NAFLD), cardiovascular diseases, cancer, musculoskeletal diseases (e.g., arthritis), sleep apnea, respiratory conditions (such as asthma), and others. MetS is characterized by specific pathologic conditions, including obesity, insulin resistance, hypertension, and hyperlipidemia [4]. NAFLD presents liver steatosis in the absence of alcohol abuse and is closely linked to metabolic syndrome [5]. Currently, NAFLD is the most important cause of chronic liver disease, ranging from simple steatosis to severe tissue inflammation (steatohepatitis), fibrosis, cirrhosis, and hepatocellular carcinoma [6]. 

Treatment options for obesity are based on lifestyle changes (healthy diet, physical exercise, and behavioral therapies), pharmacotherapy, and bariatric surgery. Pharmacotherapy, allied with exercise and diet, is a standard support for obesity treatment, although some drugs have been considered non-efficacious, unsafe, and showed dangerous side effects [7,8]. Due to the exposed facts, the development of more specific, effective, and directed treatments and drugs against obesity and obesity-related pathologies is still in need. Natural products are known as a source of novel molecules for treating human diseases, and a significant portion of the approved drugs (> 60%) has origin in nature or inspired the synthesis of pharmacophores. One of the FDA-approved drugs for the treatment of obesity is derived from the bacterium *Streptomyces* (orlistat), which targets the inhibition of gastric and pancreatic lipases [3]. 

Researchers have been focusing on terrestrial environments for new natural compounds discovery for centuries, while research has only started looking at aquatic environments in the last few decades. Despite the apparent difficulties of isolating novel compounds from marine organisms in a sustainable way, currently, 14 FDA-approved drugs from marine environments exist for the treatment of cancer, pain, or hypertriglyceridemia (https://www.midwestern.edu/departments/marinepharmacology/clinical-pipeline, accessed on 6 December 2021). Oceans comprise more than 71% of the Earth’s surface and represent a huge potential reservoir for discovering new natural compounds [9]. Microalgae are known to be a source of lipids, proteins, and carbohydrates [10,11], as well as being rich in exploitable molecules, vitamins, polyunsaturated fatty acids (PUFAs), pigments, carotenoids, other antioxidant compounds, peptides, and toxins [12,13]. The macronutrients produced vary depending on the microalgae species and its cultivation mode (autotrophic, heterotrophic, and mixotrophic) [13]. As a source of high-value products and molecules, microalgae became interesting for commercial purposes, such as biotechnological and pharmaceutical exploitation, biodiesel production, and food and dietary supplements, among others. 

The microalgae *Chlorella vulgaris*, the first algae isolated and grown in pure culture is often used for bioremediation and treatment of wastewater [14,15], and is approved for production and commercialization as a health/food supplement. The regular dietary supplementation of *C. vulgaris* can protect against specific types of cancer, oxidative stress and help in the reduction of hyperlipidemia [16]. Recent studies have demonstrated *C. vulgaris’s* ability to produce carotenoids, chlorophyll pigments, a range of fatty acids and complex lipids (glycerophospholipids, glycolipids) [17], and vitamins (B3 and B5), among other interesting compounds [18]. In contrast to the commonly cultivated green algae *C. vulgaris*, *Chlorococcum* sp. is a green microalga from the family Chlorococcaceae, which is still poorly explored in terms of its biotechnological potential. *Chlorococcum* sp. is known for accumulating large amounts of lipids [19] and producing high amounts of carotenoids [20]. Furthermore, *Chlorococcum* sp. has anticholinesterase and antioxidant activity, and the presence of phenolic compounds, polysaccharides, and omega 3 polar lipids [21] was shown by Olasehinde et al. [22], making this microalga a possible interesting nutraceutical producer.

The present work aimed to search for novel bioactivities from microalgae with human health applications, particularly for metabolic diseases. This work focussed on two microalgae, *Chlorococcum amblystomatis* grown autotrophically and *Chlorella vulgaris* grown auto- and heterotrophically, and their biomass was extracted using organic solvents. Those extracts were evaluated in various in vivo and in vitro bioassays for their activities towards obesity, steatosis, and inflammation. Additionally, potential toxicity was assessed, and the first characterization of their metabolite profile by untargeted LC-MS/MS was performed to identify putative compounds involved in the observed bioactivities. The identification of bioactive extracts for metabolic diseases from microalgae and their responsible metabolites will be an important step for the future development of novel nutraceuticals or functional food.

## 2. Results

### 2.1. Lipid-Reducing Activity

Microalgae extracts were subjected to a bioactivity screening to identify their potential lipid-reducing activities using zebrafish larvae in vivo, namely the Nile Red fat metabolism assay. Results are represented in Figure 1. Dimethyl sulfoxide (0.1% DMSO) was used as solvent control and resveratrol (50 µM REV) as a positive control. Figure 1a–b shows a visualization of Nile Red staining of neutral lipid reservoirs for the solvent control and the positive control, respectively. Results from bioactivity screening are presented in Figure 1c and expressed as a percentage of fluorescence intensity relative to the solvent control (DMSO). Seven of the analyzed samples reduced significant Nile Red fluorescence. In particular, heterotrophic *C. vulgaris* (CH), extracted with ethanol and ultrasounds (UAE), revealed very promising results, similar to the ones obtained from the positive control, REV. Even though most of the results showed an elevated variability, with large standard deviations (SD) for most samples, CH-UAE extract showed very small SD. No toxicity or visible malformation was detected during the assays, assuming the absence of toxicity in zebrafish larvae at the tested concentration. 

Comparing the heterotrophic, most active extract (CH-UAE, 75% lipid reduction) with the equivalent autotrophic extract (CA-UAE, 20% lipid reduction), it is possible to infer that the cultivation mode plays an essential role in the production of bioactive compounds present in the extracts.

### 2.2. Anti-Inflammatory Activity

The anti-inflammatory activity was assessed using the RAW264.7 cell line exposed to 10 and 25 μg/mL extracts. Results from 10 μg/mL are presented in the Appendix A. Figure 2 shows the results from the extracts at 25 μg/mL. Eight of the nine samples resulted in statistically significant nitric oxide (NO) reduction, ranging from approximately 25% to more than a 40% reduction in NO content (Figure 2a). Analyzing Figure 2b, which represents viability, no cytotoxicity was observed for extracts at 25 µg/mL extracts. In contrast, one extract significantly increased viability (CA-DM; 42% average viability increase). 

### 2.3. Anti-Steatosis Assay

The steatosis assay quantifies the lipid content through the fluorescence emitted by the Nile Red dye, which stains neutral lipid droplets, and cell viability through the intensity of fluorescence emitted by HO-33342 that stains the nucleus (Figure 3a). Results presented in Figure 3b consists of a ratio between the intensity emitted by the Nile Red per cytoplasm and the intensity emitted by the HO-33342, giving the results as the lipid content normalized for cell density for each well. At 10 µg/mL, no significant results were obtained (Appendix A). At 25 µg/mL, six of the nine extracts showed slight, but significant anti-steatosis activity, when compared to the fat-overloaded control (control + SO). Like the results obtained from the zebrafish Nile Red assay, ultrasound-assisted extraction with ethanol in heterotrophic *C. vulgaris* (CH-UAE) showed a 20% reduction in lipid content in HepG2 cells, reinforcing this as the most promising extract. Results observed in the SrB assay (Figure 3c) discarded significant cytotoxicity for all samples.

### 2.4. Metabolite Profiling

For the untargeted metabolite profiling, the extracts were chosen and compared, forming three groups: autotrophic *Chlorella vulgaris* (CA-DM; CA-E; CA-UAE), heterotrophic *Chlorella vulgaris* (CH-DM; CH-E; CH-UAE), and *Chlorococcum amblystomatis* (C-DM; C-E; C-UAE). The molecular network (Figure 4) allowed the visualization of the metabolites present in each group or shared between groups, their precursor mass, and putative identifications (http://gnps.ucsd.edu/ProteoSAFe/status.jsp?task=9f0f25a1f005406eab16a5e5711a19ea, accessed on 25th of October 2021). The original Cytoscape file is provided in Appendix A. 

Most molecules are shared between all three groups (white nodes) or between both autotrophically grown microalgae (orange nodes). For example, although chlorophylls are shared by all three groups, some chlorophyll metabolites are exclusive to *C. amblystomatis* extracts (red nodes) or autotrophic growth (orange nodes). Betaine lipids and terpene lactones metabolites were only detected in autotrophic growing conditions. On the other hand, we were able to identify a cluster of triglycerides only identified in heterotrophically grown *C. vulgaris*. These results suggest that culture conditions may play an important role for the production of the metabolites present in the extracts, even more than the species itself. Metabolites shared with both *C. vulgaris* groups and colored in green are scarce in the molecular network. 

Metabolites only present in CH-UAE extract, identified in the previous assays as the most promising ones, were highlighted in Figure 4 in blue and octagonal nodes. From this extract, 18 unique nodes in clusters were identified. Table 1 shows the results obtained for those 18 nodes, with their putative identifications. 

## 3. Discussion

The use of cell lines for bioactivity screening in metabolic diseases, as obesity and associated co-morbidities, have proved to be an essential tool; however, they do not represent the complexity of whole organisms. Zebrafish (*Danio rerio*) surged as an attractive model organism. It has high physiological and genetic homology with mammals, relatively low-cost maintenance, larvae with optical transparency, and fast development. Similarities between zebrafish and mammalian lipid metabolism have been recorded [23], allowing the development of specific screening assays. The zebrafish Nile Red fat metabolism assay [23] allows the quantification of the neutral lipidic content (in the intestine and yolk sac region) in zebrafish larvae, enabling us to understand if certain compounds have lipid-reducing activity. The zebrafish Nile Red assay has been successfully used for screening of new secondary metabolites from cyanobacterial fractions [24] and for the isolation of known and novel chlorophyll derivatives (13^2^-hydroxypheophytine a, 13^2^-hydroxy-farnesyle a) from the cyanobacteria *Nodosilinea* sp. and *Cyanobium* sp. [25].

To the best of our knowledge, this assay was not yet employed in microalgae extracts, and we report, for the first time, strong bioactivity for heterotrophic *Chlorella vulgaris* extracted with ethanol and ultrasound-assisted extraction, which reduced 70% of neutral lipids in the zebrafish larvae. In comparison, autotrophic *C. vulgaris* only reduced 20% of neutral lipid levels. In concordance with these results, *C. vulgaris* anti-obesity activity has long been identified and studied [26,27,28]. In high-fat-diet fed rats, supplementation with *C. vulgaris* effectively reduced total serum lipids, liver triglycerides, and cholesterol [28].

Various authors revealed differences between autotrophic and heterotrophic cultivation. Chen et al. [29] had demonstrated that the lipid content and microalgae composition are species- and cultivation-dependent. Heterotrophic growth is more profitable in terms of lipid content when compared to traditional autotrophic growth conditions [30,31]. Although both culture conditions were closed systems with a more controlled environment, the heterotrophic conditions are axenic, allowing an easier manipulation and production of more constant biomass while decreasing the probability of culture contamination [32]. Autotrophic cultivation is described to lead to a higher content of glycolipids and omega-3 polyunsaturated fatty acids, whereas heterotrophic growth increased phospholipids, saturated fatty acids, and omega-6 polyunsaturated fatty acid levels [17]. Autotrophic cells were shown to have a higher concentration of pigments (chlorophyll and carotenoid), while heterotrophic cells have lower pigment content [33].

Molecular networking allowed further investigation of putative compounds associated with the bioactivity previously identified in the heterotrophic extract from *C. vulgaris* (CH-UAE). The comparison between microalgae and growth conditions revealed unique metabolites in this bioactive extract. Of those presented in Table 1, none was yet described in the scientific literature for their lipid-reducing activity. Additionally, such compounds’ production is not associated with microalgae but with bacteria, plants, or fungi, in accordance with the information in the databases. The other 12 peaks with no match in the searched databases may represent novel compounds. In accordance, many studies using similar untargeted metabolite profiling approaches demonstrated a large percentage of not-known compounds, which still remain to be isolated and identified [24,34,35,36]. In the future, CH-UAE extract should be fractionated to reduce the complexity of the material and identify the responsible compounds for the observed anti-obesity activity.

As obesity is characterized by chronic inflammation, it is essential to assess the anti-inflammatory activity of the extracts. Our results showed a reduction in the production of NO of about 30–40% for *C. vulgaris* and *C. amblystomatis*. Couto et al. [17], when comparing both autotrophic and heterotrophic *C. vulgaris,* were able to conclude that both growing conditions had high anti-inflammatory and antioxidant properties. However, results in that study were obtained using 500 μg/ml extracts, which is 20x higher than those used in the present study (25 μg/mL). The production of bioactive compounds with anti-inflammatory activity has already been reported in *C. vulgaris* [37] and *Chlorococcum* sp. [22]. A study from Kwak et al. [38] showed anti-inflammatory and immunostimulatory effects through an increase in natural killer cells activity and the production of cytokines in healthy people after *C. vulgaris* biomass consumption for an eight-week period. The carotenoid violaxanthin from *Chlorella ellipsoidea* had been associated with anti-inflammatory activity [39], capable of inhibiting NO production in a dose-dependent way [39].

Steatosis, one of the NAFLD states, is common in obese patients [40]. The assay employed here to determine anti-steatosis activity relies on the importance of the liver in fatty assay metabolism and used HepG2 cell lines exposed to sodium oleate, known to induce steatosis in cells, as described by Costa et al. [24]. Although both microalgae under different growth conditions demonstrated significant results, there are differences in these activities. Extraction with dichloromethane-methanol 2:1 (DM) and ultrasound-assisted extraction with ethanol (UAE) resulted in higher anti-steatosis activity compared with ethanol (E) extracts. The effect of ultrasound-assisted extraction, when compared to extraction just with ethanol, was also observed at the anti-inflammatory assay. Ethanol extracts from *Coffea arabica* L. leaves have been characterized to be more efficient for extracting chlorophylls, carotenoids, and higher antioxidant activity when compared to dichloromethane or hexane extracts [41]. Solvent mixtures, compared to single solvents, were found to be more efficient, with higher yields in extracts of *Annona muricata* L. leaves [42]. Ultrasound extraction presents a variety of advantages, such as lower time consumption, as well as improved extraction yield for organic compounds, compared to conventional methods [43,44,45].

Regarding *C. amblystomatis* extracts, not much research has been reported on lipid-reducing activity in in vitro or in vivo models of obesity or steatosis. This microalga is not accepted for human consumption, and therefore cannot be subjected to human trials, which may explain the lack of further studies. Our results for *C. amblystomatis* extracts are promising, as it shows a significant reduction in neutral lipids in fatty-acid-overloaded liver cells compared with the ones obtained for heterotrophic *Chlorella*. In accordance with the zebrafish Nile Red fat metabolism assay results, ultrasound-assisted ethanolic extract of heterotrophic *C. vulgaris* demonstrated the highest bioactivity in preventing the formation of lipid droplets in HepG2 cells. Our study allows inferring that both microalgae species have a lipid-lowering effect that may be useful for the treatment of NAFLD.

## 4. Materials and Methods

### 4.1. Microalgae Biomass Production

*Chlorella vulgaris* and *Chlorococcum amblystomatis* are deposited at Allmicroalgae industrial collection under the internal codes 0002CA and 0066CA, respectively. These microalgae were cultivated as previously described [17,46]. Briefly, autotrophic cultivation was carried using Guillard f/2-based medium. The scale-up started in 5 L flask reactors under laboratory-controlled conditions, which were sequentially scaled up, at an approximate proportion of 1:5 until reaching an outdoor 10 m^3^ photobioreactor (PBR) at Allmicroalgae production plant facilities. The pH was kept constant by pulse injections of CO_2_. For heterotrophic cultivation, a C:N ratio of 6.7:1 and glucose (Sapecquimica, Vila Nova de Gaia, Portugal) was used as the source of organic carbon. The heterotrophic culture was carried step-wise from 50 mL Erlenmeyer’s until reaching the industrial 5000 L fermenter. All fermenters were operated in fed-batch under controlled temperature, pH, and dissolved oxygen.

### 4.2. Microalgae Extraction

Microalgae biomass extraction was performed using 25 mg of *Chlorella vulgaris* (grown autotrophically and heterotrophically) and *Chlorococcum amblystomatis* using ethanol 96% (E), dichloromethane:methanol 2:1 (DM), and ultrasound-assisted extraction (UAE) with ethanol 96%. The UAE was performed using a Sonics VCX 130 sonifier (Sonics & Materials INC., Newtown, CT., USA, output power 130 W, output frequency 20 kHz, power density 3.56 W/cm^3^), with a microtip probe set to six 20 second pulses of 70% of amplitude, each followed by one minute cool down in ice, as described in Figueiredo et al. [47]. Samples were vortexed for 2 min and centrifuged at 2000 rpm for 10 min. The organic phases were collected and dried under a N_2_ stream.

Crude microalgae extracts were weighed in pre-weighed amber vials. For bioactivity assays, dry crude extracts were diluted in DMSO at a concentration of 10 mg/mL.

### 4.3. Zebrafish Larvae Nile Red Fat Metabolism Assay

The lipid-reducing activity of compounds was analyzed using the zebrafish Nile Red assay as previously described [8]. According to the EC Directive 86/609/EEC for animal experiments, the chosen procedures are not considered animal experiments using non-autonomous feeding stages, and no permission was necessary. Zebrafish embryos were raised from one DPF (days post-fertilization) in egg water (60 μg/mL marine sea salt dissolved in distilled H_2_O) with 200 μM PTU (1-phenyl-2-thiourea) to inhibit pigmentation. From three DPF to five DPF, zebrafish larvae were exposed to the samples at a final concentration of 10 μg/mL with daily renewal of water and extracts in a 48-well plate with a density of 6–8 larvae/well (*n* = 6–8). A solvent control (0.1% DMSO) and positive control (REV, resveratrol, final concentration 50 μM) were included in the assay. Neutral lipids were stained with Nile Red overnight at the final concentration of 10 ng/mL. For imaging, the larvae were anesthetized with tricaine (MS-222, 0.03%) for 5 min and fluorescence analyzed with a fluorescence microscope (Olympus, BX-41, Hamburg, Germany). Fluorescence intensity was quantified in individual zebrafish larvae by ImageJ (http://rsb.info.nih.gov/ij/index.html, accessed on 1 July 2021).

### 4.4. Cell Assays

The murine macrophage cell line RAW 264.7 (American Type Culture Collection, ATCC) was selected to determine the anti-inflammatory potential. RAW 264.7 were cultured in Dulbecco’s Modified Eagle Medium (DMEM, Roti-CELL) with glutamine, without pyruvate, supplemented with 10% (*v*/*v*) of inactivated fetal bovine serum (FBS) and 1% (*v*/*v*) penicillin-streptomycin (penicillin 100 IU/L, streptomycin 100 μL/mL). Human-hepatoma-derived cell line HepG2 cells (ATCC) were used for the anti-steatosis assay. Cells were cultured in Dulbecco Modified Eagle’s Medium (DMEM) and grown in DMEM supplemented with 10% (*v*/*v*) fetal bovine serum, 1% penicillin/streptomycin (100 IU mL^−1^ and 10 mg mL^−1^, respectively), and 0.1% amphotericin. Both cell lines were incubated in a humidified atmosphere with 5% CO_2_ at 37 °C. The culture medium was renewed twice a week, and cell passages (scraping for RAW 264.7 and trypsinization for HepG2) were made at about 80% confluence. All samples were tested at a final concentration of 10 µg/mL.

#### 4.4.1. Anti-Inflammatory Assay

The anti-inflammatory assay was performed as described by Lopes et al. [48]. RAW 264.7 cells were stimulated with LPS (1 μg/mL) and incubated for 22 h. After the incubation period, NO was measured in the culture medium through a Griess reaction. A total of 75 μL of Griess reagent (sulfanilamide 10 mg/mL and ethylenediamine 1 mg/mL, prepared in 2% H_3_PO_4_) was mixed with 75 μL cell supernatant and incubated in the dark for 10 min. The absorbance of the reaction product was measured at 562 nm. Results were expressed as the percentage of NO from the LPS control. Three independent assays were performed, each assay in duplicate for each sample. To assess the direct effect of the extracts on basal NO production, the assay was also performed in RAW 264.7 cells in the absence of LPS (pro-inflammatory activity).

Cytotoxicity of the extracts was monitored through the 3-(4,5-dimethylthiazole-2-yl)-2,5-diphenyltetrazolium bromide (MTT) assay, as described by Lopes et al. [48]. The assay consisted of the reduction in the yellow MTT to insoluble purple formazan crystals by dehydrogenizing metabolically active cells. After the incubation period of 24 h, 100 μL of MTT solution (0.5 mg/mL), freshly prepared in DMEM at 37 °C, was added to each well and incubated at 37 °C for 45 min. The supernatant was removed after the incubation period, and the resulting formazan crystals were dissolved in 100 μL DMSO. The absorbance of the colored product was determined at 515 nm. Cytotoxicity was expressed as the percentage of cell viability vs. the solvent control. Three independent assays were performed in duplicate for each sample.

#### 4.4.2. Steatosis Assay

The anti-steatosis assay was performed as described by Costa et al. [24]. Cells were seeded at 10^5^ cells/mL in 96-well plates and adhered overnight (24 h). The cells were washed in PBS and changed to incomplete DMEM supplemented with 62 μM sodium oleate. DMSO and 0.5% MeOH were used as solvent control. Cells were incubated at 37 °C for 6 h. After, cells were changed to Hank’s Buffered Salt Solution (HBSS) (0.137 M NaCl, 5.4 mM KCl, 0.25 mM Na_2_HPO_4_, 0.44 mM KH_2_PO_4_, 1.3 mM CaCl_2_, 1.0 mM MgSO_4_, 4.2 mM NaHCO_3_, glucose-free) with HO-333424 (1:100) and Nile Red5 (1:400), and incubated for 15 min at 37 °C in the dark. Cells were then washed twice with HBSS. Fluorescence was read at 485/572 nm excitation/emission for Nile Red and 360/460 nm for HO-33342.

Cytotoxicity of the fractions was also tested on HepG2 cell line using the SRB colorimetric assay. Following the anti-steatosis assay, cells were fixed for 1 h at 4 °C, in the dark, adding 50% (*w*/*v*) ice-cold trichloroacetic acid (TCA) to the culture medium. Cells were washed four times with deionized water and the plates air-dried. Then, 0.4% (*w*/*v*) SRB in 1% acetic acid was added to each well for 15 min, followed by five washes with 1% acetic acid. The plates were air-dried, and 10 mmol L−1, pH 10.5 Tris–HCl was added to each well. Absorbance was read at 492 nm with reference at 650 nm on a Synergy HT Multi-detection microplate reader.

### 4.5. Metabolite Profiling

A total of 40 µl of each sample replica (DMSO, 10 mg/mL) was dried and then resuspended to a concentration of 1 mg/mL in acetonitrile for LC-MS. Experimental conditions are described in Ribeiro et al. [49]. The chromatographic step was carried out in an ACE UltraCore 2.5 Super C18 column (75 mm × 2.1 mm, Advanced Chromatography Technologies, Aberdeen, United Kingdom). Briefly, mobile phase A was a mixture of water (H_2_O; 95%), methanol (MeOH; 5%), and formic acid (0.1%); while mobile phase B consisted of a mixture of isopropanol (95%), MeOH (5%) and formic acid (0.1%). The gradient flux was 0.35 mL/min, and the program ran for 20 min. The separation temperature was kept at 40 °C for the entire analysis. Q Exactive™ Focus Hybrid Quadrupole carried out the LC/MS analysis—Orbitrap™ Mass Spectrometer (Thermo Scientific™, Waltham, MA, USA) coupled to an Electrospray Ionization (ESI) source, operating in positive mode.

As described by Bellver et al. [34], molecular networking analysis was conducted with some modifications. Raw data were converted using the software program MSConvert to mzXML format for molecular networking and metabolomic analysis. Converted data were uploaded to GNPS (Global Natural Products Social Molecular Networking) [50] (https://gnps.ucsd.edu/, accessed on 25 October 2021). A molecular network was created using the online workflow (https://ccms-ucsd.github.io/GNPSDocumentation/, accessed on 29 October 2021) on the GNPS website (http://gnps.ucsd.edu, accessed on 25 October 2016) [50]. The data were filtered by removing all MS/MS fragment ions within +/− 17 Da of the precursor m/z. MS/MS spectra were window filtered by choosing only the top 6 fragment ions in the +/- 50Da window throughout the spectrum. The precursor ion mass tolerance was set to 0.02 Da and an MS/MS fragment ion tolerance of 0.02 Da. A network was then created where edges were filtered to have a cosine score above 0.7 and more than 6 matched peaks. Further, edges between two nodes were kept in the network if and only if each of the nodes appeared in each other’s respective top 10 most similar nodes. Finally, the maximum size of a molecular family was set to 100, and the lowest-scoring edges were removed from molecular families until the molecular family size was below this threshold. The spectra in the network were then searched against GNPS’ spectral libraries. The library spectra were filtered in the same manner as the input data. All matches kept between network spectra and library spectra were required to have a score above 0.7 and at least 6 matched peaks. Bioinformatic tools of GNPS were also used, including the Dereplicator [51], Dereplicator+ [52]. To enhance chemical structural information within the molecular network, information from in silico structure annotations from GNPS Library Search, Dereplicator, were incorporated into the network using the GNPS MolNetEnhancer workflow (https://ccms-ucsd.github.io/GNPSDocumentation/molnetenhancer/, accessed on 29 October 2021) on the GNPS website (http://gnps.ucsd.edu, accessed on 29 Octoner 2021) [50,51]. Chemical class annotations were performed using the ClassyFire chemical ontology [53]. Precursor mass was searched in GNPS as well as in other databases, such as the “Dictionary of Marine Natural Compounds” [54] (https://dnp.chemnetbase.com/, accessed on 29 October 2021) and the “Natural Products Atlas” [55] (https://www.npatlas.org/, accessed on 28 October 2021), applying a search filter of 0.002 m/z and a deviation of <5 ppm. Bioactive peaks were manually checked in the Xcalibur software (version 4.1, Thermo Scientific Exactive Series 2.9) for peak intensity, H-isotopes, and Na^+^ adducts. The MS-error was calculated in parts per million (ppm) and restricted to <5 ppm.

The obtained results from the GNPS platform were then visualized and analyzed using the software program Cytoscape 3.4.0 [56]. Nodes were grouped with different colors, and the ones with significant bioactivity were highlighted in octagonal shape and bigger size.

### 4.6. Statistical Analysis

Data were first analyzed using normality, Shapiro–Wilk, and Kolmogorov–Smirnov tests to check for Gaussian distribution of samples and a Bartlett’s test (*p* < 0.05) to determine equal variance of the samples. If the normal distribution was confirmed, a one-way analysis of variance (ANOVA) was used to find differences among means, followed by a multi-comparisons Dunnett test (*p* < 0.05) as post hoc test. If there were no normal distribution, Kruskal–Wallis test for nonparametric distribution of values would be used, using Dunn’s post hoc test. Analyses were performed using GraphPad Prism 9.0.1.

## 5. Conclusions

In conclusion, the present study shows that microalgae species, namely, *Chlorella vulgaris* and *Chlorococcum amblystomatis,* have significant anti-obesity, anti-steatosis, and anti-inflammatory activities and could be valuable resources for the development of future nutraceuticals. In particular, heterotrophic cultivation of *C. vulgaris* strongly increased the lipid-reducing activity in the zebrafish assay, confirming that alteration of culture conditions can be a valuable approach to increase the production of high-value compounds in microalgae.

## Figures and Tables

**Figure 1 marinedrugs-20-00009-f001:**
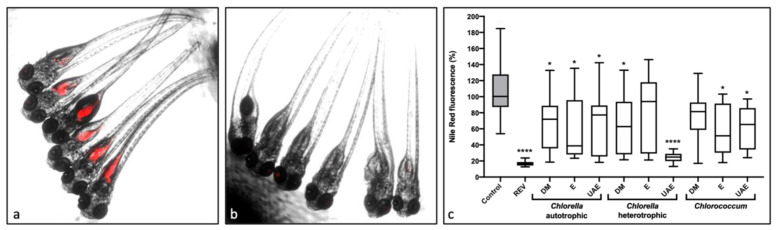
Lipid-reducing activity of the extracts (10 μg/mL) in the Nile Red fat metabolism assay using zebrafish larvae. a–b: images of zebrafish larvae, with overlay of bright field picture and red fluorescence channel; (**a**): solvent control with 0.1% of DMSO; (**b**): positive control with 50 μM of REV. (**c**): Results of the screening for lipid-reducing activity. Data are shown relative to DMSO (100%) as box-and-whisker plots (5–95 percentiles) and were obtained from three independent assays with 6–7 individual larvae each (*n* = 18–21). Statistically significant different results from control (DMSO) were marked with asterisk (* *p* < 0.05; **** *p* < 0.0001). Solvents used for preparation of the extracts: DM: dichloromethane-methanol (2:1); E: ethanol; UAE: ultrasound-assisted extraction with ethanol.

**Figure 2 marinedrugs-20-00009-f002:**
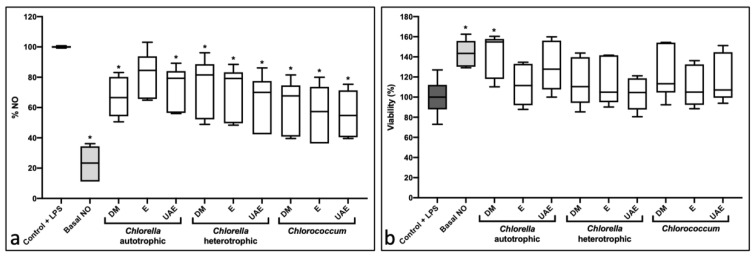
Anti-inflammatory and viability assays in RAW264.7 cell line exposed to 25 μg/mL of extracts. Dark grey represents inflammation control (induced by LPS and containing the same DMSO content as extracts); light grey represents basal NO without induction of inflammation by LPS. (**a**) Results for the anti-inflammatory assay; (**b**) results for cell viability. Solvents used for preparation of the extracts: DM: dichloromethane-methanol (2:1); E: ethanol; UAE: ultrasound-assisted extraction with ethanol. The data for both assays are derived from three independent experiments in duplicates and shown as box-and-whisker plots (5–95 percentiles). Statistical differences compared to DMSO control are indicated by asterisks (* *p* < 0.05).

**Figure 3 marinedrugs-20-00009-f003:**
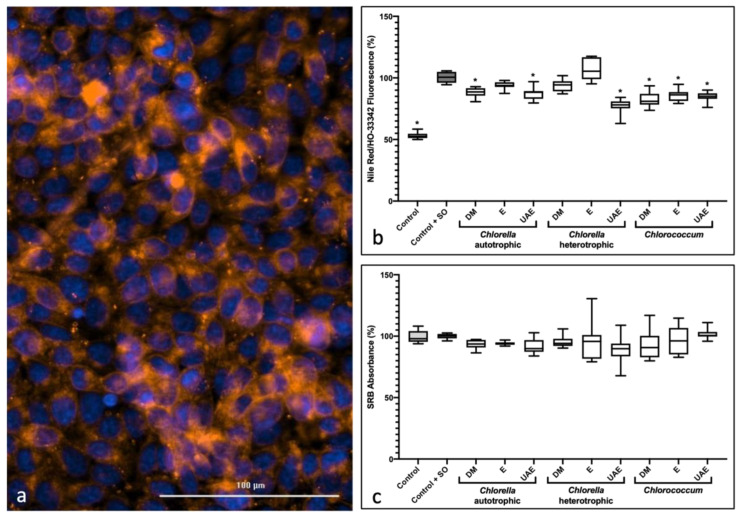
Anti-steatosis activity assay in fatty-acid-overloaded HepG2 cells and cell viability by SrB method, at 25 μg/mL extract. (**a**) HepG2 cells stained under fluorescent light; in orange, lipidic content stained by NileRed; in blue, cell nucleus stained by HO-33342 (DAPI). (**b**) Nile Red and HO-33342 fluorescence quantification ratio expressed as percentage compared to fat-overloaded control (Control + SO); (**c**) HepG2 cell viability using SrB method. Light grey represents DMSO control; dark grey represents control + SO. Solvents used for preparation of the extracts: DM: dichloromethane-methanol (2:1); E: ethanol; UAE: ultrasound-assisted extraction with ethanol. Data were derived from two independent experiments in triplicates and shown as box-and-whisker plots (5–95 percentiles). Statistical differences compared to DMSO + SO control is indicated by asterisks (* *p* < 0.05).

**Figure 4 marinedrugs-20-00009-f004:**
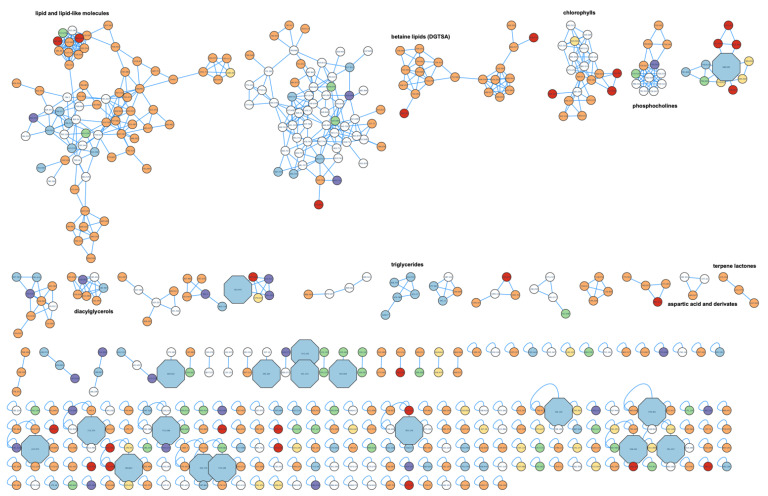
Metabolite profiling using LC-MS/MS and GNPS. Unique mass peaks, exclusively present in the most bioactive extract CH-UAE, are presented as octagonal nodes and highlighted by size. The color of the nodes corresponds to their presence in the analyzed extracts: yellow: only in autotrophic *C. vulgaris*; blue: only in heterotrophic *C. vulgaris*; red: only in *C. amblystomatis*; green: shared between auto- and heterotrophic *C. vulgaris*; orange: shared between autotrophic *C. vulgaris* and *C. amblystomatis*; purple: shared between heterotrophic *C. vulgaris* and *C. amblystomatis*; white: shared by all.

**Table 1 marinedrugs-20-00009-t001:** Putative identification of unique compounds in the active extract CH-UAE derived from the molecular network at Figure 4, by GNPS tools, DNP, and NPA. Identifications were based on the MS2 fragmentation on GNPS and on m/z values +/− 0.002 against the databases DNP and NPA. Possible matches were only considered if the calculated mass error was lower than 5ppm. From the original 18 compounds, 8 putative identifications were found. M + H^+^: mass + hydrogen; RT: retention time; ppm: parts per million; DNP: Dictionary of natural products; NPA: natural products atlas.

M + H^+^	RT	Putative Identification	ppm	Formula	Source
358.202	519.548	Benzanoid			GNPS
409.162	549.5865	2,6-Diamino-2,6-dideoxyidose; L-form, Dibenzyl dithioacetal or 3-(4-Hydroxybenzyl)-3,6-bis(methylthio)-2,5-piperazinedione; (3*R*,6*R*)-form, *O*-(3-Methyl-2-butenyl), 1,4-*N*-di-Me	0.1	C_20_H_28_N_2_O_3_S_2_	DNP
		Urauchimycin C	2.2	C_19_H_24_N_2_O_8_	DNP
333.136	749.917	Anhydrodehydrotylophorinidine; 3-*O*-De-Me	−1.5	C_21_H_18_NO_3_	DNP
		Pandangolide 2; Me ester	−3.6	C_15_H_24_O_6_S	DNP
		Xanthine; 7*H*-form, 1,7-Dibenzyl	2.5	C_19_H_16_N_4_O_2_	DNP
393.167	652.9033	7,8-Dihydroxy-1-methyl-β-carboline; 3,4-Dihydro, *O*^7^-Me, 8-*O*-β-D-Glucopyranoside	2.1	C_19_H_24_N_2_O_7_	DNP
749.391	550.2385	Biscarpamontamine A or Conodiparine A; 19’-Ketone or Conodiparine B; 19’-Ketone or Conodirinine A or Conodirinine B or Coryzeylamine or Tabercorymine A or Tabernaricatine B; 19*R*,20*S*-Epoxide or Tabernaricatine B; 19*S*,20*R*-Epoxide or Tabernaricatine D; Δ^1’,2’^-Isomer, 7’β-hydroxy	−0.6	C_44_H_52_N_4_O_7_	DNP
451.119	735.559	Aspergillazine B or Aspergillazine B; 2-Epimer	3.3	C_20_H_22_N_2_O_8_S	DNP
		2,2’,3,3’,7,7’-Hexahydroxy-1,1’-biphenanthrene or 2,2’,4,4’,7,7’-Hexahydroxy-1,1’-biphenanthrene or 2,2’,4,4’,7,7’-Hexahydroxy-1,3’-biphenanthrene or 2,4,4’,5,5’,7’-Hexahydroxy-1,1’-biphenanthrene or 3,3’,4,4’,7,7’-Hexahydroxy-1,1’-biphenanthrene or 2,4,4’,7,7’-Pentahydroxy-1,2’-biphenanthrene ether or 2,4,5’,7,7’-Pentahydroxy-1,2’-biphenanthrene ether	1.8	C_28_H_18_O_6_	DNP
		Rhizoferrin; (*R,R*)-form, 2-Oxo	−2.3	C_16_H_22_N_2_O_13_	DNP
		Aspergillazine C or Penispirozine C ou Perispirozine D	−3.3	C_20_H_22_N_2_O_8_S	NPA
729.368	732.944500	2,15-Dihydroxy-18-nor-16-kauren-19-oic acid; (*ent*-2α,15β)-form, 2-*O*-[β-D-Glucopyranosyl-(1→3)-2-*O*-(3-methylbutanoyl)-β-D-glucopyranoside] or 3,5,11,14-Tetrahydroxycard-20(22)-enolide; (3β,5β,11α,14β)-form, 3-*O*-[3-*O*-Methyl-β-D-glucopyranosyl-(1→4)-6-deoxy-α-L-glucopyranoside]	−2.4	C_36_H_56_O_15_	DNP
227.075	648.9525	3-Buten-1-ol; 4-Methylbenzenesulfonyl or 3-Buten-1-ol; 4-Methylbenzenesulfonyl or 2,4-Dihydroxy-3,5,6-trimethylthiobenzoic acid; *S*-Me ester or 4-Phenyl-3-buten-1-ol; (*Z*)-form, Methanesulfonyl	3.6	C_11_H_14_O_3_S	DNP
		1-(2’,4’-dihydroxy-5’-methyl-3’-methylsulfanylmethylphenyl)ethanone or Mortivinacin A	3.4	C_11_H_14_O_3_S	NPA
666.062	979.23				
543.447	887.89275				
415.142	734.2515				
743.346	590.652499				
402.176	711.7333				
160.841	387.5715				
761.357	554.2535				
763.178	827.886				
715.388	650.252				
713.373	775.4403				

## Data Availability

All data are contained within the article and Appendix A.

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
