# Peer review of "Potential Anti-Obesity, Anti-Steatosis, and Anti-Inflammatory Properties of Extracts from the Microalgae Chlorella vulgaris and Chlorococcum amblystomatis under Different Growth Conditions"

_marinedrugs, 2021, doi:10.3390/md20010009_

Round 1

Reviewer 1 Report

Dear authors.

Algae and microalgae attract the attention of researchers, including from the point of view on components with biological activity. The topic touched upon in the article is relevant. The scientific content of the manuscript justifies its publication. This is a good-written work. I propose to accept it for publication.

The effect of autotrophically grown Chlorella vulgaris and Chlorococcum amblystomatis extracts and heterotrophically grown Chlorella Vulgaris extract was studied. These extracts were evaluated for their bioactive properties, toxicity, and metabolite profile. The hypolipidemic effect of microalgae extracts obtained using organic solvents and ultrasound was proven.

This study's overall goal and novelty were to find new microalgae products for the prevention and treatment of obesity, particularly in metabolic diseases.

The section Materials and research methods is described in detail. The described techniques are validated and easy to reproduce based on the details provided in this section.

The Materials section is written succinctly and consistently. The tables and figures are correct and clearly describe the presented data. It is easy to interpret and understand the obtained results using tables and figures. There is an objective, easily reproducible, and understandable statistical analysis of the data.

Discussion allows assessing the importance of research results. The conditions and interpretation of experiments on zebrafish larvae are indicated.

The prospects of using the obtained results for producing fundamentally new drugs for the prevention and treatment of obesity in humans using microalgae extracts, rich in biologically active substances and possessing antioxidant, antimicrobial activity, are characterized by the absence of toxic effects on the RAW264.7 cell lines.

The manuscript is clear and well structured. The conclusion is consistent with the presented evidence and arguments. The references are relevant (about 75% were published within the past 5 years), the number of self-citations is adequate. 

Author Response

Dear authors.

Algae and microalgae attract the attention of researchers, including from the point of view on components with biological activity. The topic touched upon in the article is relevant. The scientific content of the manuscript justifies its publication. This is a good-written work. I propose to accept it for publication.

The effect of autotrophically grown Chlorella vulgaris and Chlorococcum amblystomatis extracts and heterotrophically grown Chlorella Vulgaris extract was studied. These extracts were evaluated for their bioactive properties, toxicity, and metabolite profile. The hypolipidemic effect of microalgae extracts obtained using organic solvents and ultrasound was proven.

This study's overall goal and novelty were to find new microalgae products for the prevention and treatment of obesity, particularly in metabolic diseases.

The section Materials and research methods is described in detail. The described techniques are validated and easy to reproduce based on the details provided in this section.

The Materials section is written succinctly and consistently. The tables and figures are correct and clearly describe the presented data. It is easy to interpret and understand the obtained results using tables and figures. There is an objective, easily reproducible, and understandable statistical analysis of the data.

Discussion allows assessing the importance of research results. The conditions and interpretation of experiments on zebrafish larvae are indicated.

The prospects of using the obtained results for producing fundamentally new drugs for the prevention and treatment of obesity in humans using microalgae extracts, rich in biologically active substances and possessing antioxidant, antimicrobial activity, are characterized by the absence of toxic effects on the RAW264.7 cell lines.

The manuscript is clear and well structured. The conclusion is consistent with the presented evidence and arguments. The references are relevant (about 75% were published within the past 5 years), the number of self-citations is adequate.

CIIMAR: We thank the reviewer for this opinion and time.

Reviewer 2 Report

The authors are using extracts made from microalgae and describing some of the impacts of the extracts. The authors make broad statements describing the extracts activities as anti-inflammatory, anti-steatosis, and lipid reducing.

My comments:

  1. The broad statements made about the extracts in the title should be minimized. The word extract isn’t used in the title and should be considered as some of the data are suggestive and not confirmatory. For example, one assay does not fully support something is anti-inflammatory. Using the word “potential” in the title would be more accurate.
  2. I don’t agree with the calculation used to identify the “n” used in the analysis for each graph. The n should indicate each biological repeat (done with completely separate setup), not each replicate. When an experiment done 3 different times and each time has 2 repeats the n would be 3 not 6. An experiment done twice each time in triplicate, the n would be 2 not 6. This matters for standard error calculations and is important because sometimes you see an effect on one day but not another in biological systems. I would potentially count each zebrafish as a separate n because it is a whole organism and that would carry additional sources of variability.
  3. I don’t agree with the statement that the extracts are anti-inflammatory given a decrease in NO because there is variability in the viability. The n is not labelled here in Fig 2b. How many times was the MTT assay done?. Percent viability should be set to the basal NO not the Control with LPS.
  4. The LPS used as a control shouldn’t impact viability but it does here in Fig 2b. A lower concentration of LPS should have been used.
  5. In table 1 – which extracts did these come from? Were they present in all extracts?

Author Response

The authors are using extracts made from microalgae and describing some of the impacts of the extracts. The authors make broad statements describing the extracts activities as anti-inflammatory, anti-steatosis, and lipid reducing.

My comments:

The broad statements made about the extracts in the title should be minimized. The word extract isn’t used in the title and should be considered as some of the data are suggestive and not confirmatory. For example, one assay does not fully support something is anti-inflammatory. Using the word “potential” in the title would be more accurate.

CIIMAR: We agree and have changed the title accordingly.

I don’t agree with the calculation used to identify the “n” used in the analysis for each graph. The n should indicate each biological repeat (done with completely separate setup), not each replicate. When an experiment done 3 different times and each time has 2 repeats the n would be 3 not 6. An experiment done twice each time in triplicate, the n would be 2 not 6. This matters for standard error calculations and is important because sometimes you see an effect on one day but not another in biological systems. I would potentially count each zebrafish as a separate n because it is a whole organism and that would carry additional sources of variability.

CIIMAR: We refer with “n” to the number of biological replicates, and not to the number of experimental repeats (= independent experiments). The number of replicates corresponds to biological replicates. In bioassays with cells in vitro, the biological replicates are usually in different wells of the microplate, and hence corresponds to the number of wells per condition. In zebrafish assays in vivo, the biological replicates are usually the number of individual fish per condition. Furthermore, the bioassays were performed at least in two independent experiments, which ensures the repeatability and consistency of the results. This procedure for measuring bioactivities (at least two independent experiments, and n > 6 = more than 6 biological replicates) is the standard procedure for assays in cells or in zebrafish, and can be found in countless publications in various peer-reviewed journals including Marine Drugs (and in our own previous publications, some of them published in Marine Drugs). 

I don’t agree with the statement that the extracts are anti-inflammatory given a decrease in NO because there is variability in the viability. The n is not labelled here in Fig 2b. How many times was the MTT assay done?. Percent viability should be set to the basal NO not the Control with LPS. The LPS used as a control shouldn’t impact viability but it does here in Fig 2b. A lower concentration of LPS should have been used.

CIIMAR: The number of biological replicates (n) of the anti-inflammatory assay and viability analysis is exactly the same, since it was measured from the same wells (as indicated in M&M). Supernatant is used for NO quantification, while adherent cells are used for MTT assay. However, we have complemented the information in the figure legend for clarity.

The group of comparison for the anti-inflammatory assays are the macrophage cells stimulated by LPS. The co-exposure of LPS and extracts reveal the capacity of extracts to reduce the induced inflammation by LPS. The comparison should refer to LPS stimulated conditions (percentage set to 100%), in order to understand the effects of extracts to reduce the LPS stimulated inflammation. The non-stimulated control (without addition of LPS) shows the sensitivity of the assay and that indeed LPS stimulated the production of NO (approximately 4fold). In accordance, the viability analysis should compare all extracts with the corresponding LPS control (same conditions = all groups were stimulated by LPS). No reduction of viability is observed between the LPS stimulated control and extracts, which demonstrate that extracts did not adversely affect macrophage viability.

In table 1 – which extracts did these come from? Were they present in all extracts?

CIIMAR: The table 1 is the result of the metabolite profiling. As outlined at lines 180-183, all extracts were considered for the GNPS network from autotrophic Chlorella, heterotrophic Chlorella, and autotrophic Chlorococcum. The table 1 only highlights the metabolites, which are unique to the bioactive extract, CH-UAE (Chlorella heterotrophic, ultrasound assisted extraction) as outlined in the text, the figure legend 4 and the table legend 1. We have updated the table legend for clarity.

Reviewer 3 Report

Introduction started with good emphasis on obesity, however, after 3rd paragraph a big jump to new natural compounds with an emphasis on aquatic environment is hard to swallow. Instead, convincing reason for current search for new natural compounds needs to be established and then why to research on aquatic environment has to be elevated.

One paragraph on the first algae Chlorella vulgaris is convincing, however, in the next paragraph sudden jump to Chlorococcum amblystomatis is not at all convincing. Why to focus on Chlorococcum amblystomatis?

Overall, introduction needs substantial revision. Try including the research hypothesis along with the research objectives as well.

Line 100: remove “in the present study”

Lines 100-105: move to Materials and Methods

Section 2, results are succent and convincing.

“ultrasounds” extraction is not a good usage, change it appropriately throughout the manuscript.

In the discussion mention of “Chlorococcum amblystomatis” is not seen, instead most of the discussion was centered around “Chlorella vulgaris”

Author Response

Introduction started with good emphasis on obesity, however, after 3rd paragraph a big jump to new natural compounds with an emphasis on aquatic environment is hard to swallow. Instead, convincing reason for current search for new natural compounds needs to be established and then why to research on aquatic environment has to be elevated.

CIIMAR: As requested, the introduction was updated and corrected.

One paragraph on the first algae Chlorella vulgaris is convincing, however, in the next paragraph sudden jump to Chlorococcum amblystomatis is not at all convincing. Why to focus on Chlorococcum amblystomatis?

CIIMAR: As requested, we have updated this paragraph and added such information.

Overall, introduction needs substantial revision. Try including the research hypothesis along with the research objectives as well.

CIIMAR: As requested, we have updated the section with the research hypothesis and objectives.

Line 100: remove “in the present study”

CIIMAR: Removed as suggested.

Lines 100-105: move to Materials and Methods

CIIMAR: We removed the paragraph. It was already present in similar form at M&M, and we originally included before the results in order to enhance the readability.

Section 2, results are succent and convincing.

“ultrasounds” extraction is not a good usage, change it appropriately throughout the manuscript.

CIIMAR: According to the reviewer suggestion, we have changed all into “ultrasound-assisted extraction” along the manuscript.

In the discussion mention of “Chlorococcum amblystomatis” is not seen, instead most of the discussion was centered around “Chlorella vulgaris”

CIIMAR: The discussion was centered on Chlorella vulgaris due to the fact that there is many research work available on Chlorella in the scientific literature. This enabled to compare our results to the outcomes from other studies. In contrast, few studies are yet available on Chlorococcum sp and its effects on metabolic diseases as obesity, steatosis or inflammation. Where available, such information is present in the discussion (e.g. anti-inflammatory activity).   

Round 2

Reviewer 2 Report

I appreciate that the authors have updated their title to better reflect their data.

The authors are insisting to use the n including duplicates or triplicates done in the same plate but a different well as biological replicates but they are not (even if you have published this way before does not make it correct). Duplicates or triplicates are technical replicates not biological replicates, they can be used provided they are described correctly and analyzed as such. They should not be used in the calculation of the n. The authors need to change the n or repeat the experiments.

Author Response

I appreciate that the authors have updated their title to better reflect their data.

The authors are insisting to use the n including duplicates or triplicates done in the same plate but a different well as biological replicates but they are not (even if you have published this way before does not make it correct). Duplicates or triplicates are technical replicates not biological replicates, they can be used provided they are described correctly and analyzed as such. They should not be used in the calculation of the n. The authors need to change the n or repeat the experiments.

CIIMAR: Technical replicates are defined as tests performed on the same sample multiple times. In contrast, biological replicates are defined as tests performed on biologically distinct samples representing an identical time point or treatment dose.

I think we agree that if we inject a mouse with a drug, and we would measure a biochemical parameter in the blood 3x times, we would have 3 technical replicates from the same biological sample. In order to have three biological replicates, we would need 3 different mice.

Focusing on our data:

we quantify neutral lipid reservoirs in individual zebrafish larvae, which are coming from different biological samples, regardless if we expose them in the same well or in different wells. To ensure data quality, 3 independent experiments were performed with each 6-7 individual fish resulting in 18-21 biological replicates.

Cell culture from cell lines in vitro has indeed a low biological variation, since the cell lines are clonal and basically are the same cells. If we seed cells in different wells or different plates, we assume to have different biological replicates. Technical replicates would mean if we measure the same parameter multiple times from the same well. For example, if we would use the cells for RNA isolation, and would measure a certain mRNA expression in real-time PCR using the cDNA from the same well two times in the PCR (= technical duplicates). Biological replicates would mean to measure the gene expression in different biological samples, coming from different wells. This difference becomes more evident if we think on primary hepatocytes isolated from a fish or a rodent. Primary hepatocyte culture is accepted as 3R model to reduce the number of animals in scientific experiments. The isolated cells can be seeded into multiwell plates and each well counts as a replicate. Usually accepted is to use hepatocyte cultures from 2 different animals with several replicates (for example triplicates) to ensure the consistency and repeatability of data. In order to obtain a n = 6, it is not needed to sacrifice 6 animals.

Summarizing, we do not agree with the opinion of the reviewer. However, we have adapted the figure legends, and indicated in the revised version only the number of experimental repeats and the number of replicates used per experiment.  We hope that the reviewer accepts this solution.